# Shell density of planktonic foraminifera and pteropod species *Limacina helicina* in the Barents Sea: Relation to ontogeny and water chemistry

**Siri Ofstad**[1]*, **Katarzyna Zamelczyk**[1], **Katsunori Kimoto**[2], **Melissa Chierici**[3], **Agneta Fransson**[4], **Tine Lander Rasmussen**[1]

**1** CAGE–Centre for Arctic Gas Hydrate, Environment and Climate, Department of Geosciences, UiT, The Arctic University of Norway, Tromsø, Norway, **2** Japan Agency for Marine-Earth Science and Technology (JAMSTEC), Yokosuka, Japan, **3** Institute of Marine Research, Fram Centre, Tromsø, Norway, **4** Norwegian Polar Institute, Fram Centre, Tromsø, Norway

\* siri.ofstad@uit.no

**Data Availability Statement:** The CTD and carbonate chemistry data from the crater area in June 2016 is available at Norwegian Marine Data

## Abstract

Planktonic calcifiers, the foraminiferal species *Neogloboquadrina pachyderma* and *Turborotalita quinqueloba*, and the thecosome pteropod *Limacina helicina* from plankton tows and surface sediments from the northern Barents Sea were studied to assess how shell density varies with depth habitat and ontogenetic processes. The shells were measured using X-ray microcomputed tomography (XMCT) scanning and compared to the physical and chemical properties of the water column including the carbonate chemistry and calcium carbonate saturation of calcite and aragonite. Both living *L. helicina* and *N. pachyderma* increased in shell density from the surface to 300 m water depth. *Turborotalita quinqueloba* increased in shell density to 150–200 m water depth. Deeper than 150 m, *T. quinqueloba* experienced a loss of density due to internal dissolution, possibly related to gametogenesis. The shell density of recently settled (dead) specimens of planktonic foraminifera from surface sediment samples was compared to the living fauna and showed a large range of dissolution states. This dissolution was not apparent from shell-surface texture, especially for *N. pachyderma*, which tended to be both thicker and denser than *T. quinqueloba*. Dissolution lowered the shell density while the thickness of the shell remained intact. *Limacina helicina* also increase in shell size with water depth and thicken the shell apex with growth. This study demonstrates that the living fauna in this specific area from the Barents Sea did not suffer from dissolution effects. Dissolution occurred after death and after settling on the sea floor. The study also shows that biomonitoring is important for the understanding of the natural variability in shell density of calcifying zooplankton.

## 1. Introduction

The Arctic is particularly sensitive to global warming, and this warming is greatly amplified in the Barents Sea, a large and productive shelf sea bordering the Arctic Ocean [1,2]. The Barents

Center (https://doi.org/10.21335/NMDC-225800978). Abundance data for planktonic foraminifera and pteropods can be found in PANGAEA (https://doi.org/10.1594/PANGAEA.904463). All other data is available as supporting information.

**Funding:** This work was funded by the Research Council of Norway through its Centres of Excellence scheme (grant number 223259). The XMCT analysis was funded by Japan Agency for Marine-Earth Science and Technology Grants-In-Aid for Scientific Research (KAKENHI) Grant Numbers 15H05712 and 16H04961. The water chemistry sampling and analysis was funded by the Flagship research program "Ocean Acidification and effects in northern waters" within the FRAM-High North Research Centre for Climate and the Environment, funded by The Norwegian Ministry of Climate and Environment. The funders had no role in study design, data collection and analysis, decision to publish, or preparation of the manuscript.

**Competing interests:** The authors have declared that no competing interests exist.

Sea is influenced by inflow of Atlantic Water (AW) from the south and Polar Water from the Arctic Ocean in the north, making it a hydrologically dynamic region. The two water masses mix and generate the Polar Front, a zone of very high-productivity [3]. In the northern Barents Sea there has been a substantial shift in water mass properties over the past several decades [4]. The water column in the northern Barents Sea has become warmer and more saline, and stratification has weakened [4]. This shift is due to an increase of AW water transport, and an increase in temperature and salinity of the AW [5,6]. This 'Atlantification' of the water column will impact the productivity and structure of the Barents Sea ecosystems by displacing the Polar Front north-eastward, and allowing the advection of temperate species further into the Arctic domain [6–8]. A poleward shift of species in the Barents Sea has already been documented [9–11]. The large volume of warm and saline AW is also thought to be the main cause of the rapid decline of the winter sea ice cover [1].

The Barents Sea is one of the largest $CO_2$ sink areas in the Arctic region, which is mainly caused by the year-round $CO_2$ undersaturation and high biological production [12,13] despite the formation of sea-ice in winter. The Barents Sea $CO_2$ sink is predicted to double by 2065 with an associated pH decrease of up to 0.25 pH units [14]. A significant proportion of the observed $CO_2$ increase in the Barents Sea has been from the inflow of AW, which is rich in anthropogenic $CO_2$ [15]. The meltwater from sea ice or glaciers lowers the saturation state of seawater with respect to calcite ($\Omega_{Ca}$) and aragonite ($\Omega_{Ar}$), the two most common polymorphs of $CaCO_3$ formed by marine organisms [16–18]. The volume of meltwater is predicted to increase as a result of the progressing global warming [19]. Ocean acidification (OA) may lead to adverse effects on the ability of marine calcifiers to produce $CaCO_3$ shells [20].

Planktonic foraminifera (PF) and thecosome pteropods are the major calcifiers among marine zooplankton [20]. Marine calcifiers, in particular pteropods, are important prey in many marine food webs [21–24]. In addition, both PF and pteropods contribute significantly to the biological carbon pump [25–29]. Only few studies of PF and pteropod faunas for the high Arctic exists and in particular for the Barents Sea [30–32]. Planktonic foraminifera build their shells of calcite, while the polar pteropod species *Limacina helicina* build their shells of aragonite. The crystal structure of calcite is more stable than aragonite, and the tendency for the crystal structure to dissolve is linked to the $\Omega$ in the surrounding environment of the particular mineral phase. The crystal structures of aragonite and calcite are thermodynamically stable when $\Omega>1$. Both PF and *L. helicina* are sensitive to the carbonate chemistry in their environment and the extent of their calcification is commonly used as an indicator for OA [33–40]. Furthermore, due to their long sedimentary record PF shell density has been used for paleoceanographic studies of OA and atmospheric $CO_2$ [41–44].

In a previous study, we documented the seasonal variability in the distribution patterns of PF and polar pteropod *L. helicina* and their environments in the northern Barents Sea [30]. Test size and abundance of both groups increased drastically from spring to summer, and in summer there was a clearer depth zonation of the individuals, possibly related to the thermal stratification [30]. Here, we extend our analysis on PF and *L. helicina* to study the shell density of the summer population.

In OA research there are few studies with focus on how the shell density of calcareous planktonic organisms varies with ontogeny. In contrast to the pteropod *L. helicina*, PF do not perform diel vertical migration [45]. However, their shell density and depth habitat may be linked due to the possibility of ontogenetic vertical migration meaning that they descend to a deeper habitat as their life cycle progresses, likely in order to reproduce at certain water depths [46–49]. It should be noted that this concept is still disputed and is difficult to document. We thus hypothesize that the shell density of PF is related to its depth habitat in the upper water column. As PF grow and add chambers, they add layers of calcite onto the existing shell

through secondary calcification. It is unknown how the shell density of PF changes with increasing water depth. Following the assumption that calcification is linear, it will be assumed that denser shells found deeper in the water column are older.

Furthermore, processes like ontogenetic secondary calcification, gametogenic calcite addition following gametogenesis, and diagenetic encrustation will influence how well PF are preserved in the sedimentary record, which is significant for the accuracy of studies of fossil faunas. Knowledge on the natural variability in shell density across a population of calcareous planktonic organisms will improve our ability to better document biological effects of OA. In this study, we aim to show 1) the variability in shell density of the living planktonic foraminiferal species *N. pachyderma* and *T. quinqueloba* and the pteropod *L. helicina* with shell size and water depth, 2) the interspecies differences in shell density of *N. pachyderma* and *T. quinqueloba*, 3) if any changes in the observed patterns in shell density can be related to seawater carbonate chemistry, and 4) how shell density and ontogenetic processes affect the preservation of foraminifera in the surface sediments. This study is based on X-ray microcomputed tomography (XMCT) scanning of their shells. This is a pioneer study to provide the first shell density measurements of specimens of planktonic foraminifera and *Limacina helicina* from the Arctic region.

## 2. Material and methods

### 2.1 Study and sample collection

The Barents Sea is a relatively shallow continental shelf sea adjacent to the Nordic Seas and the Arctic Ocean with a mean depth of 250 m. The Bjørnøyrenna crater area (referred to in this study as the 'crater area') (74.91˚N, 27.7˚E; Fig 1) is located in relatively deep water (~340 m depth) on the northern flank of Bear Island Trough and is characterized by high levels of methane emission [50]. The Barents Sea is mainly influenced by the inflow of warm and saline Atlantic water transported in the north-eastern flowing Norwegian Atlantic current (NwAC) and the cold Arctic water transported in the East Spitsbergen current (ESC) from the north to the south [3] (Fig 1). Once the NwAC enters the Bear Island Through it splits into two branches. A substantial part of the NwAC forms a northeast flowing current, the North Cape current, which enters the southern Barents Sea, while the remainder forms the northwest flowing West Spitsbergen current (WSC).

Samples were collected onboard *R/V Helmer Hanssen* during the expedition CAGE 16–5, on June 29[th] 2016 at three stations located at 74.9˚N, 27.7˚E–27.8˚E. No sampling permission was required at this location. This is because the study area is outside of the 12-mile limit of the Norwegian coast, meaning it is not in territorial waters, and the sampling causes no harm to the environment. The plankton sampled from the water column are not endangered or protected species. The PF and *L. helicina* were sampled with a stratified plankton net with mesh size of 64 µm (net opening 0.5 m$^2$; Hydro-Bios, Kiel, Germany), from five consecutive depth intervals (0–50 m, 50–100 m, 100–150 m, 150–200 m, and 200–300 m). Parallel measurements and sampling for the study of physical and chemical environment in the water column were performed at the same location using a Conductivity-Temperature-Depth (CTD)-Rosette system with seawater sampling for determination of carbonate chemistry. Empty shells found in the water column >150 m are assumed to represent recently dead specimens. Their shells were transparent, well preserved and similar to the shells of the live specimens containing protoplasm.

### 2.2 Sampling of marine calcifiers

Once the plankton tows were retrieved, the samples were sieved with sea water through a 63-µm sieve and transferred into plastic bottles (250 ml) and fixed and buffered with

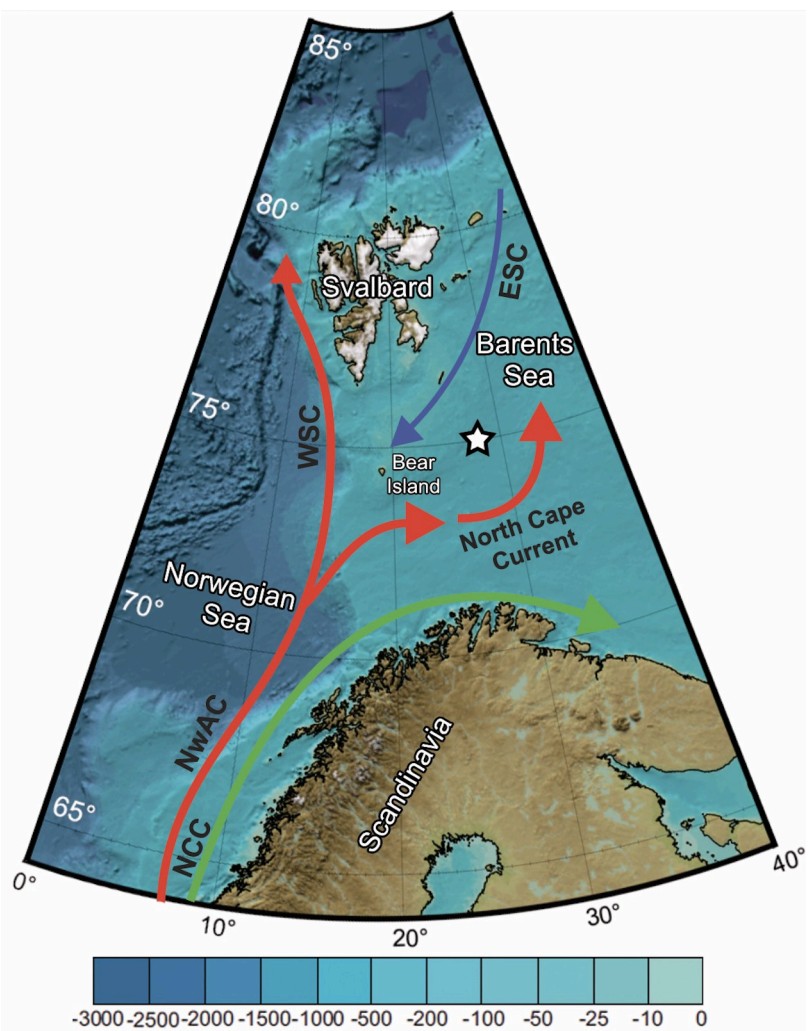

**Fig 1. Schematic map of study area and main current systems in the Nordic Seas.** White star indicates the crater area where plankton tows, box-cores and water sampling were conducted, detailed bathymetry can be found in Ofstad et al. [30]. Red lines are Atlantic Water inflows, blue line is Arctic Water outflows, and the green line is a coastal current. Abbreviations: *NwAC* Norwegian Atlantic current, *WSC* West Spitsbergen current, *ESC* East Spitsbergen current, *NCC* Norwegian Coastal Current. Current systems are based on Loeng [3]. Basemap from IBCAO 3.0 [51].

approximately 230 ml ethanol (98%), a quarter of a teaspoon hexamethylenetetramine (≥99.0%), and stored at 2˚C. Once in the laboratory, the samples were washed over a 63-µm sieve in order to remove organic particles from the surface of the foraminiferal tests and to break up aggregations of material. All PF and *L. helicina* from the >63-µm size fraction were picked with a fine brush under a light microscope. Live (cytoplasm-bearing) planktonic foraminifera specimens were counted for each depth.

Recently settled planktonic foraminifera were collected from two box-cores located within the same area as the plankton tow stations (74.92˚N, 27.77˚E and 27.53˚E). The water depth at both box-core stations was 330 m, and the $\Omega_{Ca}$ directly above the sediments was 1.22 [30]. The PF were collected by sampling the top sediment layer (1 cm) of the boxcore. The samples were preserved in approximately 50 ml of ethanol (96%) with rose bengal (2 g L$^{-1}$ of ethanol), and stored at 2˚C. In the home laboratory, the samples were washed over a 63-µm sieve and dried in a 40˚C for at least 24 hr. Once dried, PF were picked under a light microscope with a fine

brush and identified to species level. There were large pteropods in the sediment samples, but they were broken, and therefore not included in the study. The complete description of sample collection, treatment, and analysis is described in Ofstad et al. [30].

## 2.3 XMCT

An XMCT system (ScanXmate-DF160TSS105, Comscantecno Co. Ltd., Kanagawa, Japan) was used to quantify the shell density of individual specimens. A high-resolution setting (X-ray focus spot diameter of 0.8 μm, X-ray tube voltage of 80 kV, detector array size of 1024x1024 for the pteropods and 992x992 for the foraminifera, spatial resolution of 0.833 μm for *Limacina helicina* and 0.964 μm for the foraminifera, 1200 projections/360˚, 4 s/projection) was used for 3-D quantitative densitometry of the foraminiferal and pteropod tests. One to three samples (depending on the shell size) were placed on a stage made of a quartz glass bar. Tests were mounted on the sample stage with urethane glue. A calcite crystal ball was used to standardize the computed tomography (CT) number of each test sample and enabled us to distinguish the density distributions in the foraminiferal and pteropod tests with high resolution. In this study, a limestone particle (diameter of approximately 130 μm; 1000 in mean CT number; NIST RM8544 (NBS19)) was embedded in the sample stage, and all of the test samples were scanned with the same calcite standard. ConeCTexpress software (White Rabbit Corp., Tokyo, Japan) was used to correct and reconstruct tomography data, and the general principle of Feldkamp cone beam reconstruction was followed to reconstruct image cross sections based on filtered back projections. In order to avoid the beam hardening effect (selective attenuation of X-ray) during scan, we put the metal filter (Aluminium, 0.2 um thickness) in front of X-ray detector. Mean shell thickness was calculated by dividing the $CaCO_3$ volume by the shell surface area, both of which are parameters measured by the XMCT. The shell surface area includes both the outer areas and the surfaces of the internal chambers. A caveat with the calculated mean shell thickness is that values will decrease, when the shell material is more porous. High porosity of the shell material increases the surface area, resulting in a decrease in mean shell thickness.

Well-preserved specimens to be scanned with the XMCT were selected at random, but with the intention of having a representative size range. The complete size range of the PF and *L. helicina* specimens sampled in June 2016 from the crater area can be found in Ofstad et al. [30]. A total of 226 planktonic foraminifera shells from the water column (*N. pachyderma* n = 120, *T. quinqueloba* n = 115), 30 recently settled planktonic foraminifera shells (*N. pachyderma* n = 12, *T. quinqueloba* n = 18), and 25 *Limacina helicina* shells from all depth intervals (0–50 m, 50–100 m, 100–150 m, 150–200 m, and 200–300 m) were scanned with the XMCT (S1 Table in S1 File; Fig 2). All scanned pteropod shells were either veligers, *Limacina* spp. (<300 μm, n = 7), or juveniles *L. helicina* (300–4000 μm, n = 18) [52].

## 2.4 CT number

From the 3-D scanning data of planktonic foraminiferal and *L. helicina* tests, we obtained a CT number of each volumetric pixel—referred to as a voxel, and volume ($μm^3$) of each individual test. The 3-D imaging software Molcer Plus (White Rabbit Corp., version 1.35) and the following equation were used to calculate the calcite CT number:

$$\text{CT number} = [(\mu_{\text{sample}} - \mu_{\text{air}})/(\mu_{\text{calcite STD}} - \mu_{\text{air}})] \times 1000 \qquad (1)$$

where $\mu_{\text{sample}}$, $\mu_{\text{air}}$, and $\mu_{\text{calciteSTD}}$ are the X-ray attenuation coefficients of the sample, calcite, and air, respectively.

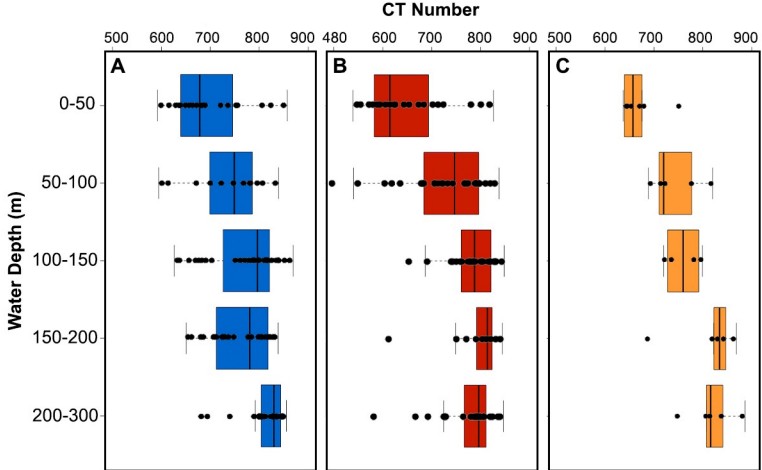

**Fig 2.** Box-and-whisker plot of shell density with water depth for A) *Neogloboquadrina pachyderma* (n = 120), B) *Turborotalita quinqueloba* (n = 115) and c) *Limacina helicina* (n = 25) sampled from the crater area in 2016. Boxes extend from the lower to upper quartile values of the data, with a line at the median. Whiskers indicate 1.5 times the inter-quartile distance. Black dots are single measurements.

The mean CT number for an entire test was calculated with the following equations:

$$\text{Mean CT number} = \frac{1}{T}\sum_{n=230}^{1000} nT_n \qquad (2)$$

where n is the CT number, $T_n$ is the total number of voxels with a specific CT number (n), and T is the total number of voxels in the whole test. The mean CT number indicates the mean density of an individual test.

## 2.5 CT data analysis

The shell thickness of the apex of *L. helicina* was measured by creating cross-sections using the Molcer Plus software (Version 1.35). A whorl is a single 360° revolution of the shell spiral structure. The shell apex of 16 shells of *L. helicina* were measured at four locations, twice on the protoconch (first whorl), and twice on the second whorl (S1 Fig). Careful consideration was made to take measurements at the same location for each shell for ease of comparison. Following the methods outlined by Janssen [53], the *L. helicina* shell diameters were measured and the total number of whorls were counted to the nearest quarter (S1 Fig). Additional *L. helicina* from the sampling station were measured for their shell diameter. Images were acquired by a Leica Z16 APO microscope, using the integrated Leica DFC450 camera and LAS version 4.12.0 software. The images were processed using the ruler tool in Adobe Photoshop CS6. All measurements of shell diameter and thickness performed this study are the result of three repeated measurements to diminish inaccuracies.

In order to calculate area density (area normalised weight), 111 PF shells (*T. quinqueloba* n = 54, *N. pachyderma* n = 57) shells were weighed individually using a Sartorius microbalance (model M2P, 0.1μg sensitivity). The given weight measurements are based on three repeated measurements of the single specimen. Area density is given by shell weight divided by surface area.

Isolation of the penultimate and final chamber was done on a select number of shells in order to validate the relationship with the overall CT number of the shell.

## 2.6 Statistical analyses

To test the relationship between any two parameters (e.g., water depth and mean shell density), a simple linear regression model was applied to the data. To test significance of correlation of shell density of the marine calcifiers with sampling intervals, a Mann-Whitney-U test was performed using Version 1.2.1335 of the program R [54]. When testing variables against water depth, the maximum depth in the plankton tow sampling intervals was used, e.g., 50 m for the sampling interval 0–50 m. When testing against environmental parameters, the mean of all measurements taken in the sampling interval was used. Typically, two water samples were taken with the CTD within a plankton tow sampling interval, once at the shallowest point, and once at the deepest. We believe using the mean of those two measurements within the sampling interval would give the most representative value.

## 2.7 Ocean carbonate chemistry

The water chemistry data were published in Ofstad et al. [30], here we give a brief overview of the methods. Dissolved inorganic carbon (DIC) was determined using gas extraction of acidified sample followed by coulometric titration and photometric detection using a Versatile Instrument for the Determination of Titration carbonate (VINDTA 3C, Marianda, Germany). Routine analyses of Certified Reference Materials (CRM, from A. G. Dickson, Scripps Institution of Oceanography, USA) ensured the accuracy and precision of the measurements. Average standard deviation from triplicate CRM analyses was within ±1 μmol kg$^{-1}$ for all samples. Total alkalinity ($A_T$) was determined from potentiometric titration with 0.1 N hydrochloric acid in a closed cell using a Versatile Instrument for the Determination of Titration Alkalinity (VINDTA, Marianda, Germany). Average standard deviation for $A_T$, determined from triplicate CRM measurements was ±2 μmol kg$^{-1}$. We used DIC, $A_T$, salinity, temperature, and depth for each sample as input parameters in a $CO_2$-chemical speciation model (CO2SYS program, version 01.05) [55,56] to calculate other parameters in the carbonate system such as carbonate-ion concentration ($[CO_3^{2-}]$), aragonite saturation ($\Omega_{Ar}$) and calcite saturation ($\Omega_{Ca}$). We used the $HSO_4^-$ dissociation constant of Dickson [57], and the $CO_2$-system dissociation constants ($K^*1$ and $K^*2$) estimated by Mehrbach et al. [58], and modified by Dickson and Millero [59].

## 3. Results

### 3.1 Hydrography and water chemistry

During the time of sampling, the predominant water masses were Atlantic Water (AW, T > 3.0°C, S > 34.65) in the top 250 m of the water column, and Transformed Atlantic Water (TAW, T = 1.0–3.0°C, S > 34.65) below 250 m, following the definitions of Cottier et al. [60] (S2 Fig). Both $\Omega_{Ar}$ and $\Omega_{Ca}$ were supersaturated ($\Omega$>1) throughout the entire water column, with the highest values in the surface water and lowest at the bottom (Figs 3D and 8B). The water column had two distinct layers (Figs 3D and 8B). The upper layer is from the sea surface to approximately 75 m water depth (S2 Fig); here, the $\Omega_{Ar}$ is 2.1–2.5, and the $\Omega_{Ca}$ is 4.0–3.0 (Figs 3D and 8B). Between 75 m and 300 m water depth the $\Omega_{Ar}$ is 1.5–2.1, and the $\Omega_{Ca}$ is 2.4– 3.0, where the lowest values were observed at the bottom (Figs 3D and 8B). The $[CO_3^{2-}]$ ranged between 168 μmol kg$^{-1}$ at the surface and 105 μmol kg$^{-1}$ at 300 m water depth. The pH ranged between 8.03 and 8.22.

### 3.2 Shell density from CT number

For both *Neogloboquadrina pachyderma* and *Turborotalita quinqueloba*, the average CT number increases steadily from 684 and 632 in the 0–50 m depth interval to 762 and 793 in the

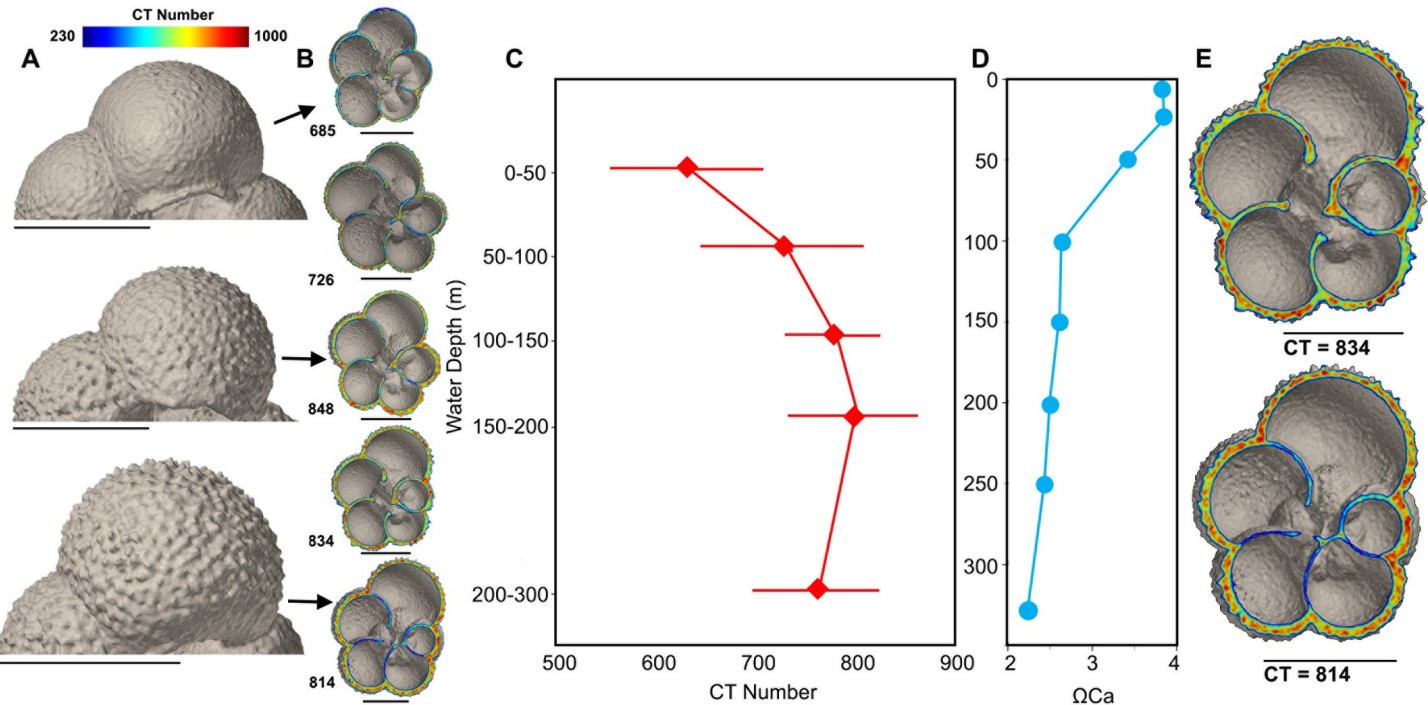

**Fig 3. *Turborotalita quinqueloba* from water column.** A) Texture of test surface of *Turborotalita quinqueloba* at three different depth intervals; 0–50 m, 100–150 m and 200–300 m. B) Variation in inner and outer shell density of *T. quinqueloba* as mean CT number of entire shell measured by XMCT increases. C) Mean CT number of *T. quinqueloba* (n = 115), with error bars, plotted against water depth. D) Calcite saturation at sampling site plotted against water depth. E) *T. quinqueloba* cross-section before and after assumed gametogenesis. Scale bars measure 100 µm.

150–200 m depth interval, respectively (Fig 2A and 2B). The difference in CT number between the layer of elevated Ω saturation at 0–50 m and the underlying water column when normalized for shell volume, is also significant for both PF species (p < 0.01), but not *L. helicina* (p = 0.25). For *L. helicina* the difference in shell density between the specimens in the shallow layer (0–50 m) and those found beneath is significant when not size normalized (S10 Table in S1 File). *Turborotalita quinqueloba* reaches its peak shell density of 793 in the 150–200 m depth interval. Below the 150–200 m depth interval, the shell density of *T. quinqueloba* decreases. In the 200–300 m depth interval, the average shell density of *T. quinqueloba* is 766. The outer shell walls are thick and dense, while the CT number is lower in the internal walls (Fig 3E and S3 Fig). In contrast, the shell density of *N. pachyderma* continues to increase until 200–300 m, where it reaches a peak shell density of, on average, 813 (Fig 4C). Similar to *N. pachyderma*, the shell density of *L. helicina* increases with depth. At 0–50 m, *L. helicina* have an average CT number of 670, and by 200–300 m they reach a peak average density of 819 (Fig 2C). Collectively, we found that the difference in shell density between sampling intervals were most significant between the shallowest (0–50 m) and deepest (200–300 m) interval (S8-S10 Tables in S1 File). *Turborotalita quinqueloba* showed the most significant variation between net tows, and *L. helicina* the least.

Although we found a general increase in CT number and shell thickness with depth, we note a large range in CT numbers (Fig 2 and S2 Table in S1 File) and mean shell thickness (S2 Table in S1 File) at each sampling depth interval. This is particularly true for *T. quinqueloba* in the shallowest depth interval 0–50 m where the CT numbers of individual specimens are evenly distributed from 539 to 826, and the mean shell thickness ranges from 2.02 to 3.25 µm. Furthermore, in the 0–50 m depth interval the average CT numbers for *N. pachyderma* and *L.*

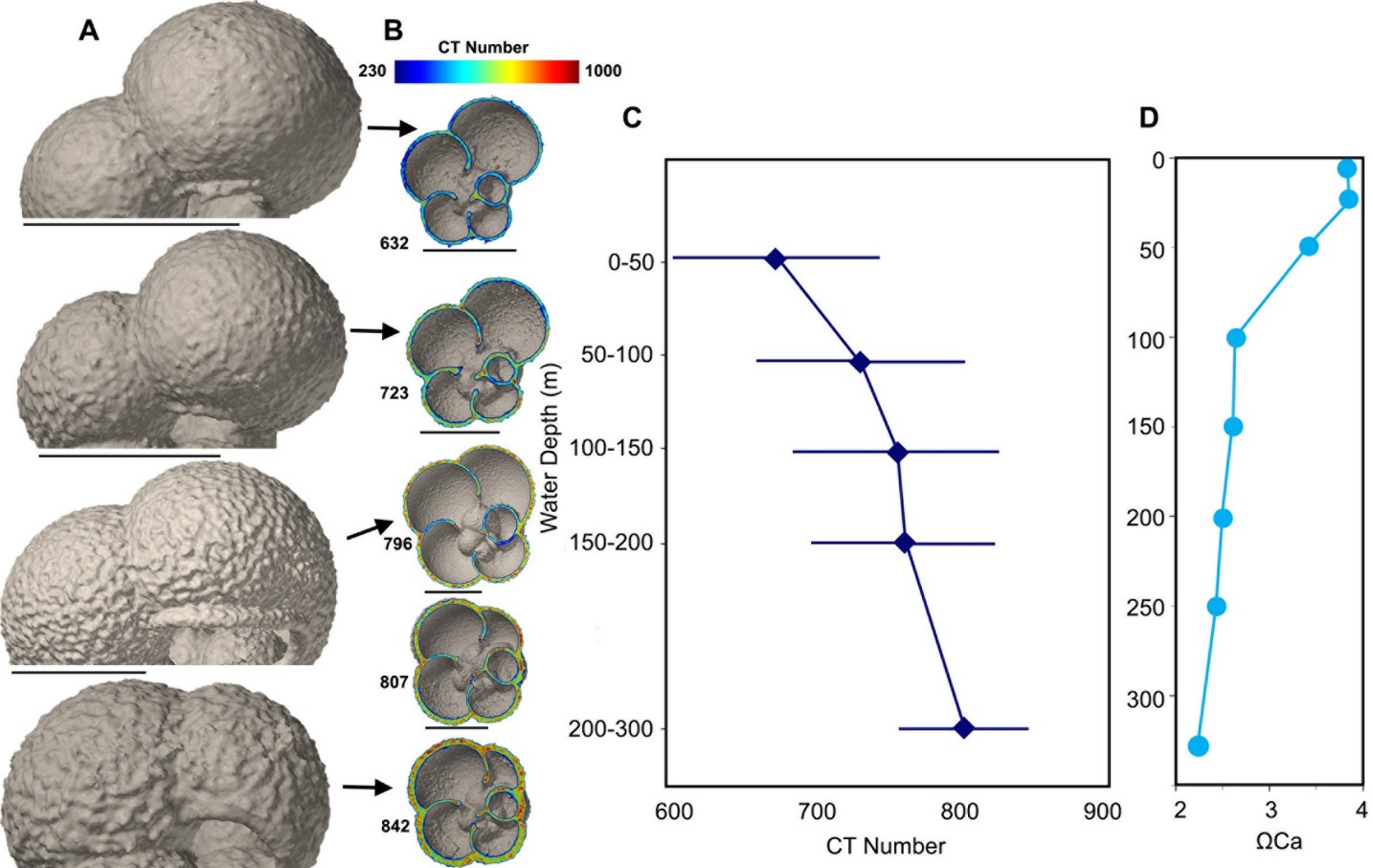

**Fig 4. *Neogloboquadrina pachyderma* from water column.** A) Texture of test surface of *Neogloboquadrina pachyderma* at four different depth intervals; 0–50 m, 50–100 m, 100–150 m and 200–300 m. B) Variation in inner and outer shell density of *N. pachyderma* with mean CT number of entire shell measured by XMCT. C) Mean CT number of *N. pachyderma* (n = 120), with error bars, plotted against water depth and calcite saturation. D) Calcite saturation at sampling site plotted against water depth. Scale bars measure 100 µm.

*helicina* ranges from 592 to 857, and 637 to 751, respectively. The shell thickness of *N. pachyderma* and *L. helicina* at the 0–50 m depth interval ranged from 1.94 to 5.28 µm, and 1.98 to 2.75 µm, respectively.

## 3.3 Planktonic foraminifera

**3.3.1 Planktonic foraminifera from the water column.** Both *N. pachyderma* and *T. quinqueloba* show a statistically significant positive correlation between individual shell weight, CT number, mean shell thickness and area density with water depth (S4, S5 Tables in S1 File). Cytoplasm-bearing specimens of both species are found in each sampling depth interval and constitute 80–100% of XMCT-scanned shells from the top 150 m (S7 Table in S1 File). Below 150 m the percentage of live specimens decreases to 75% and 78.6% for *N. pachyderma* in the 150–200 m and 200–300 m depth interval, respectively (S7 Table in S1 File). For *T. quinqueloba* there is a greater decrease in the percentage of live specimens below 150 m, with 14.3% and 23.5% containing a cytoplasm in the 150–200 m and 200–300 m depth interval, respectively (S7 Table in S1 File). For both *T. quinqueloba* and *N. pachyderma* there is increasing formation of a layer of secondary calcite crust on the outer shell with depth. The texture of the

shells in the shallowest samples are smooth without any calcite crust. Thereafter ridges appear that become increasingly "rough" with depth and increase in CT number (Figs 3A and 4A).

Both species undergo gradual shell thickening with depth. At 0–50 m water depth the average shell thickness of *N. pachyderma* and *T. quinqueloba* is 2.5±0.8 μm (n = 15) and 2±0.5 μm (n = 28), respectively. *Neogloboquadrina pachyderma* reaches peak thickness at 200–300 m, where the average shell thickness is 4.3±0.7 μm (n = 29). *Turborotalita quinqueloba* reaches peak thickness at 150–200 m, where the average shell thickness is 3.5±0.7 μm (n = 13). In the 200–300 m depth interval the shell thickness of *T. quinqueloba* has decreased to 3.1±0.8 μm (n = 24). Collectively, the shell walls of *N. pachyderma* and *T. quinqueloba* thicken by 40.8% and 35.1%, respectively, from their thinnest at the 0–50 m sampling interval to their peak shell thickness.

The mean shell thickness shows a strong correlation with the CT number (Fig 5; S4, S5 Tables in S1 File). The mean shell thickness of individual *T. quinqueloba* and *N. pachyderma* have an exponential relationship with their respective CT numbers (Fig 5). The exponential curve for *N. pachyderma* is steeper than the curve for *T. quinqueloba*. Furthermore, *N. pachyderma* (n = 120) tend to be larger, denser, and thicker than *T. quinqueloba* (n = 115), based on mean CT numbers and calcite volume (S1, S2 Tables in S1 File).

**3.3.2 Planktonic foraminifera from the surface sediments.** In the top 1 cm of the sediments, both *N. pachyderma* and *T. quinqueloba* are found in a wide range of dissolution states. Some of the planktonic foraminiferal specimens found in the surface sediments have similar shell densities as those found in the overlying water column (Figs 6A, 6C, 7A and 7C), while

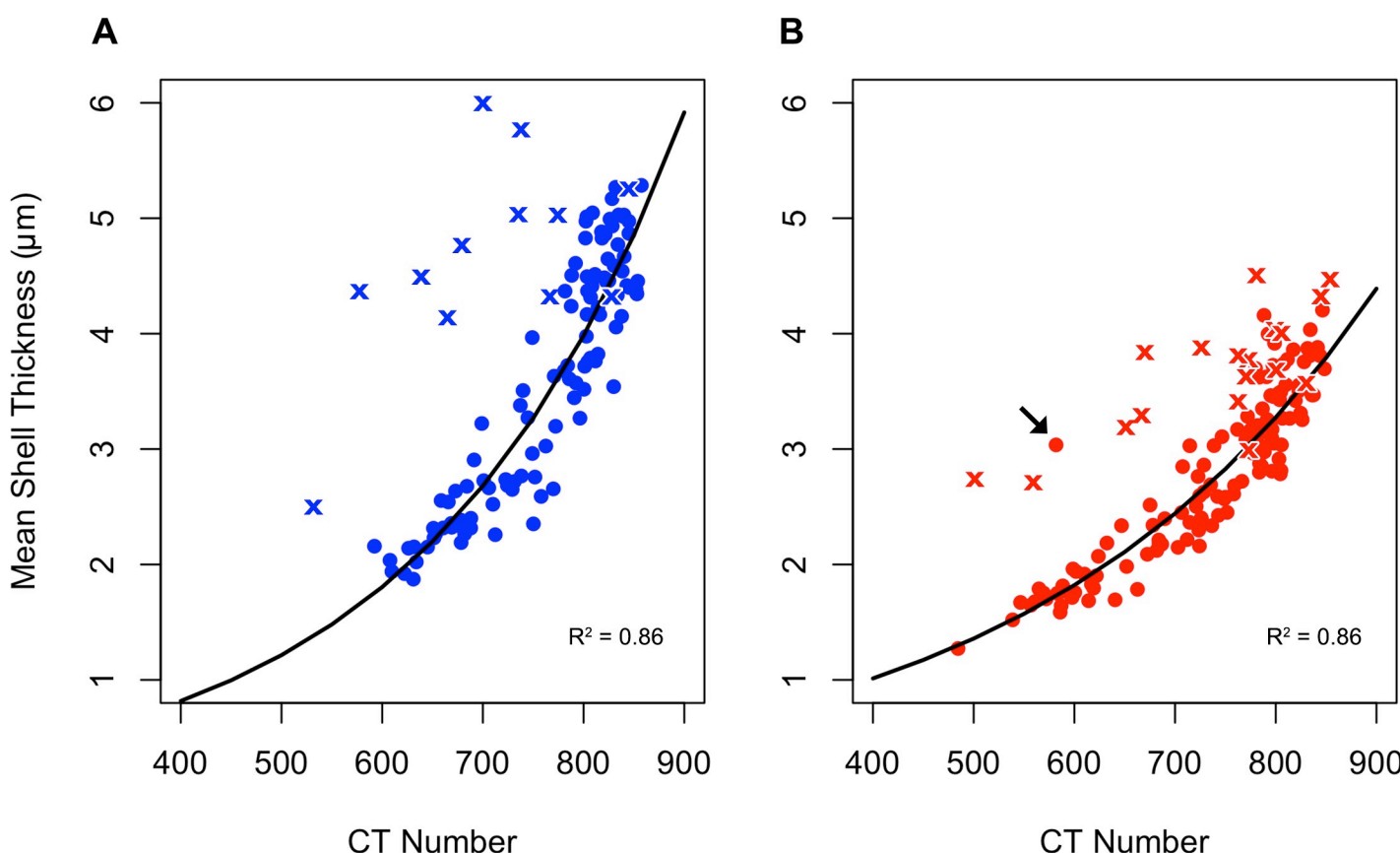

**Fig 5. Shell thickness versus shell density.** Mean shell thickness of A) *Neogloboquadrina pachyderma* and B) *Turborotalita quinqueloba* plotted versus mean shell density in the form of a CT number, fitted with an exponential model. Shells from water column samples are represented by circles, while crosses represent shells from surface sediments. Exponential model is only fitted to shells from water column. Arrow in B) is pointing to an outlier.

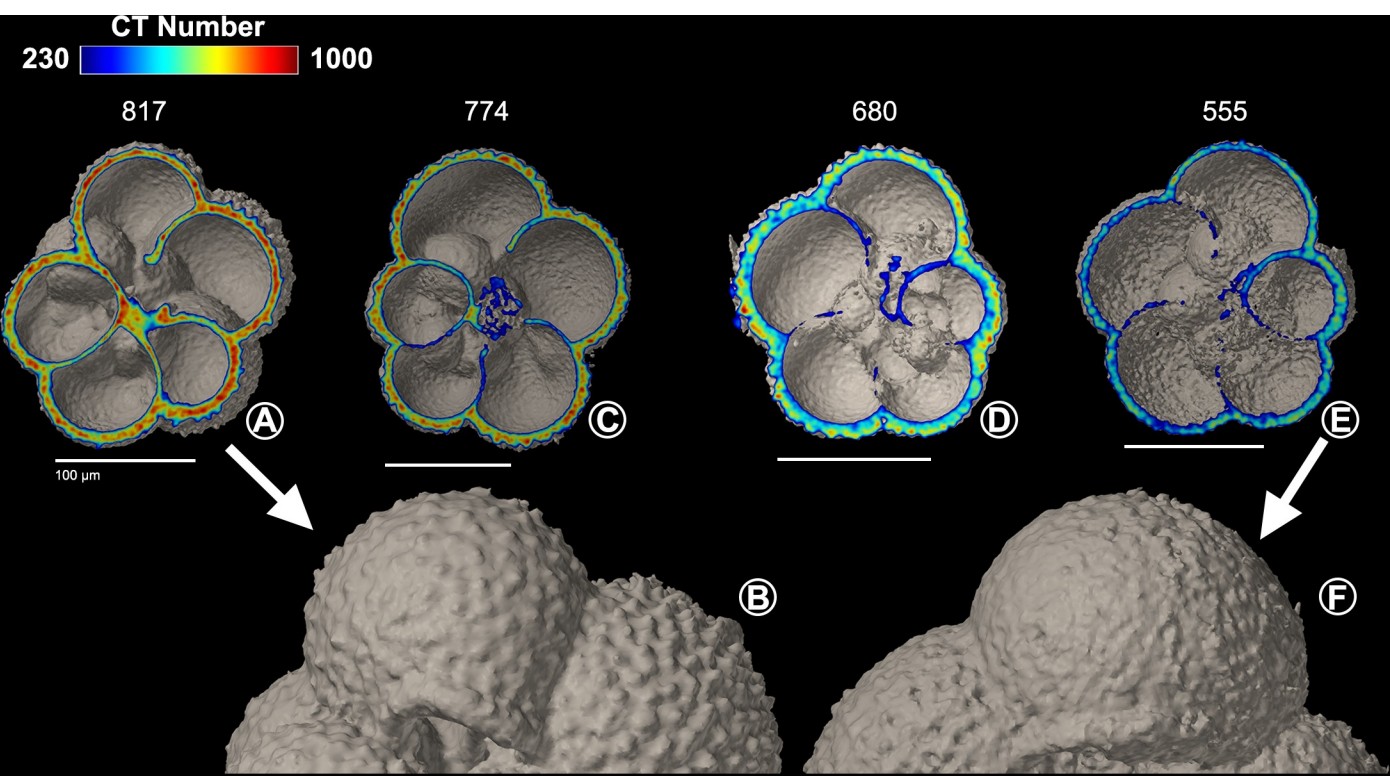

**Fig 6. *Turborotalita quinqueloba* from surface sediments.** Cross-sections of *Turborotalita quinqueloba* specimens (A,C,D,E) from surface sediment sample (0–1 cm), including surface texture of a B) high-density (n = 11) and a F) low-density specimen (n = 7). Scale bars measure 100 μm.

other specimens have undergone dissolution (Figs 6D, 6E, 7E and 7F). Out of all of the *N. pachyderma* shells found in the surface sediments, there is a high proportion of low-density shells (9 out of 12, 75%), i.e., shells which can be regarded as outliers in the thickness versus density plot (Fig 5A). In contrast, low-density *T. quinqueloba* shells are in the minority (7 out of 18, 39%) (Fig 5B). The surface texture of *N. pachyderma* and *T. quinqueloba* vary in terms of CT number (Figs 6 and 7). In *T. quinqueloba*, the loss of the base features of the prominent spines is evident as the CT number reduces from 817 to 555, and the surface texture takes on a smoother appearance (Fig 6B and 6F). The surface texture of *N. pachyderma* appears to be mostly unaffected by post-depositional dissolution (Fig 7B and 7G). In the low-density shells, the calcite ridges are more prominent, giving it a more rugose texture overall (Fig 7G). In *N. pachyderma* we see a two-layered dissolution pattern (Fig 7F). There is a clear divide between the less dense (CT number ~ 400) inner calcite, and the denser outer crust (CT number ~ 650) (Fig 7F). Shells of both species from the surface sediments that have undergone post-depositional dissolution plot to the left of the exponential trendline (Fig 5). The external shell walls of the dissolved specimens remain at a similar thickness to those with a high-density shell (Figs 5–7). Dissolution primarily affects the CT number (Fig 5).

### 3.4 *Limacina helicina*

In *L. helicina* we see the same trend in the shell density with water depth as we do with the PF (Fig 2). *Limacina helicina* show a statistically significant positive correlation between shell diameter, CT number, and mean shell thickness with water depth (S6 Table in S1 File). On average, the shell density of *L. helicina* increases with depth (Fig 8A). The mean density given

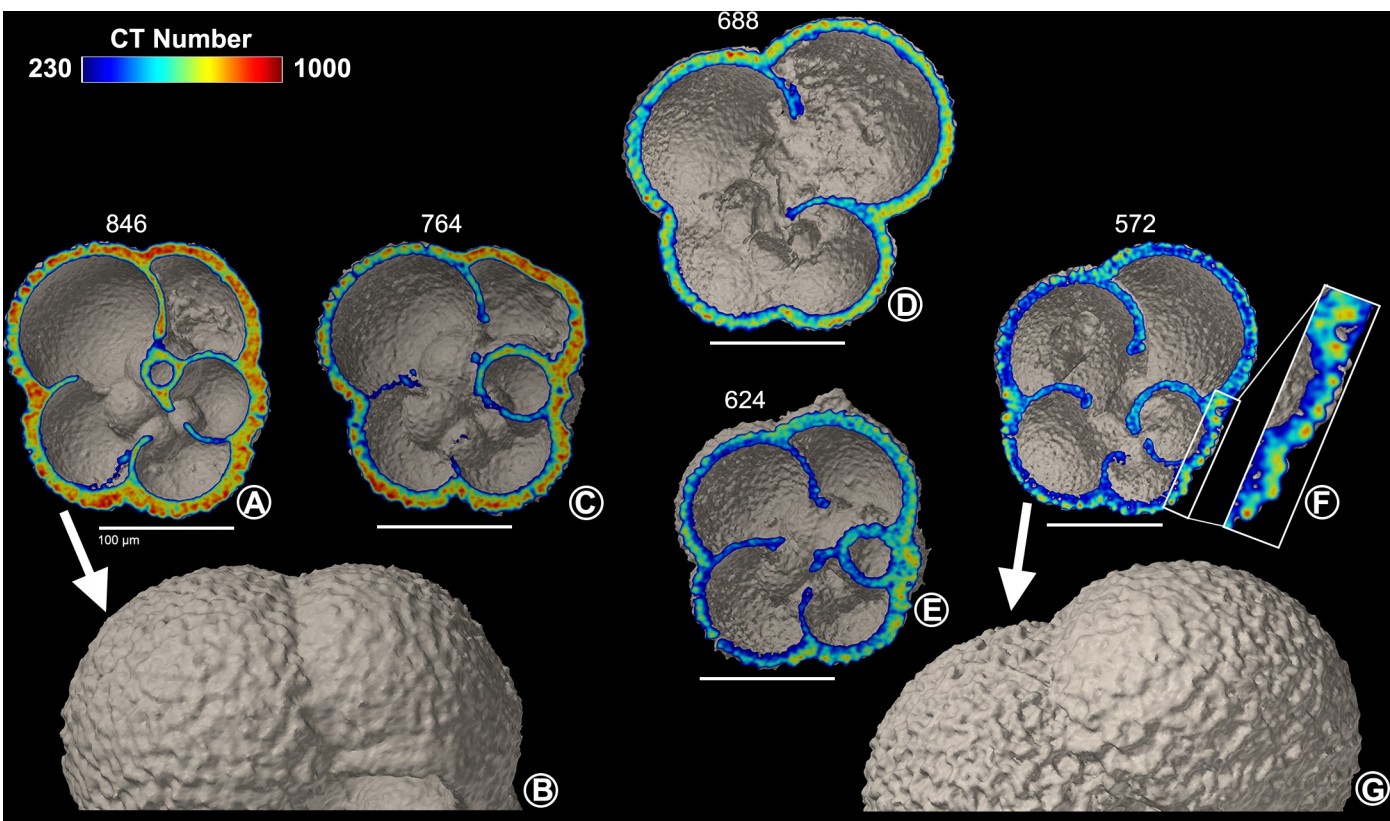

**Fig 7. *Neogloboquadrina pachyderma* from surface sediments.** Cross-sections of *Neogloboquadrina pachyderma* specimens (A,C,D,E,F) from surface sediment sample (0–1 cm), including surface texture of a B) high-density (n = 3) and a G) low-density specimen (n = 9). F) Close-up of shell wall cross-section. Scale bars measure 100 μm.

by the CT number starts at a minimum, at 670, in the shallowest sampling interval (0–50 m) (Fig 8A). There is a steady increase until the deepest sampling interval where the mean CT number is 819 (Fig 8A). In contrast to the PF in the crater area, *L. helicina* generally increase in shell diameter with depth (Fig 8A; S6 Table in S1 File). In the 0–50 m depth interval, the shells have the narrowest size range (131–457 μm), and an average size of 274 μm. The 150–200 m water depth interval has the largest range of shell sizes, 124–1190 μm (Fig 8A). The largest shells, on average, are found in the 200–300 m water depth interval and are 511 μm (Fig 8A). The number of whorls varied between 0.6 and 3.6 and is strongly correlated to the shell diameter (p < 0.001).

The distribution of *L. helicina* in the water column in terms of shell density results in an inverse relationship with $\Omega_{Ar}$ ($R^2$ = 0.54, p < 0.001, Fig 8A and 8B). The mean shell thickness also increases with depth, starting at 2.2 μm at 0–50 m water depth, to 2.8 μm at 200–300 m water depth. As the number of whorls increases, the shell apex thickens. The sum of four measurements done on the central-top part of the shell show that shells with 2.5 to 3.5 whorls is 25.9±3.1 μm, while shells with 1.5 to 2.25 whorls has a sum of 19.4±2 μm (Fig 8D).

## 4. Discussion

### 4.1 Distribution of PF, life cycles and shell density

Calcified shells are thought to have evolved as a mean for protection, and is widely found throughout the animal phyla [61]. Calcification intensity, the term often used to refer to shell

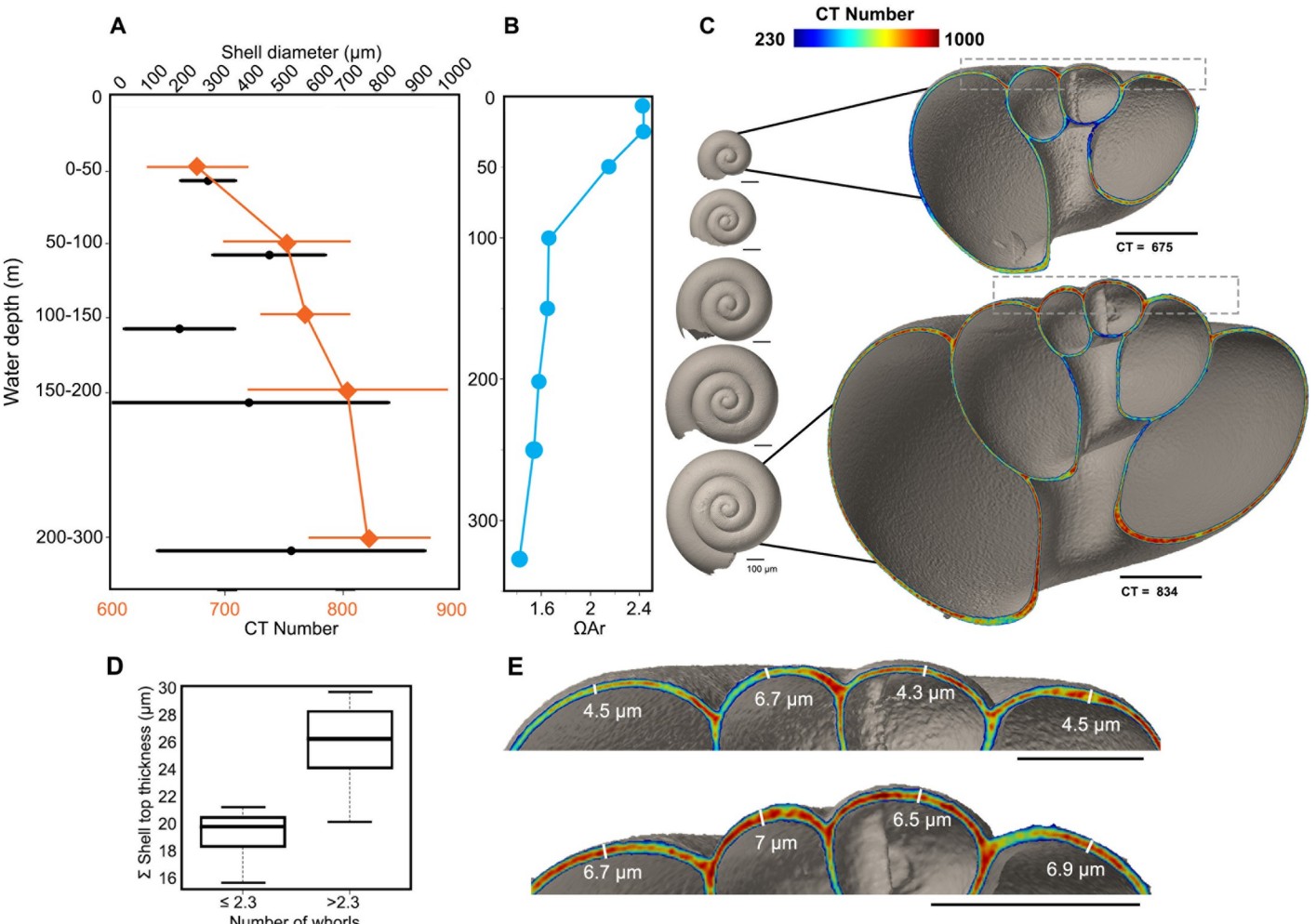

**Fig 8. *Limacina helicina* from water column.** A) *Limacina helicina* shell diameter (n = 175) and density (n = 25) (given by CT number) with depth. B) Aragonite saturation at sampling site plotted against water depth. C) Generalized shell size with depth (left) and cross-sections of *L. helicina* specimens from 0–50 m (2 whorls), and 150–200 m (2.75 whorls) water depth interval. Grey boxes are shown as close-ups in E. D) Boxplot of Mann-Whitney *U* test on top shell thickness of *L. helicina* as a function of whorl number. E) Top of *L. helicina* specimens shown in C, schematic of shell thickness measurements performed on all shells. Scale bars measure 100 μm.

density is believed to be primarily controlled by ambient seawater $[CO_3^{2-}]$ [38,62], and hence $\Omega$, which is largely dictated by absolute $[CO_3^{2-}]$. In addition, the shell size of PF appears to be controlled by temperature and food availability [36,38,63]. *Globigerina bulloides* when growing in favourable conditions, but with low $\Omega_{Ca}$ (~1.5), were found to grow large in size, with low density tests characterised by large and porous crystalline structures, suggesting that PF in some cases may prioritize shell size over shell density [36]. Furthermore, shell thickening by secondary calcification during ontogeny and/or gametogenic calcite addition is poorly understood and exhibit inter-species variation [64,65]. In polar waters, *N. pachyderma* with and without a thick calcite crust generated by secondary calcification were found to be concentrated in different parts of the water column. They were also found to add the calcite crust primarily at 50–200 m water depth, and an increase in secondary calcification of *N. pachyderma* was shown to occur with depth [66]. The degree of ontogenetic crust formation in *N. pachyderma* is highly variable,

it can amount to 50–70% of the total shell weight, and there is no consensus to which factors control the crust formation [66–68].

**4.1.1 Comparison of *N. pachyderma* and *T. quinqueloba* in the water column and their preservation patterns.** The dominant living planktonic foraminiferal species in the polar region are *N. pachyderma* and *T. quinqueloba* [69–73], which is reflected in our sampling area [30]. The differences in the shell density depth profile between *N. pachyderma* and *T. quinqueloba* can be explained in part by the differences in depth habitat and depth of reproduction (Fig 2A and 2B) [74,75]. Another factor, which may affect their calcification is that *T. quinqueloba* is a spinose species, while *N. pachyderma* is not. *Turborotalita quinqueloba* calcify within 25–75 m water depth, while *N. pachyderma* calcify within the much wider range of 25–280 m [74,75]. Our interpretation of the shell density profile is that *Neogloboquadrina pachyderma* continue to calcify and apparently grow denser as they migrate to deeper depths throughout their lifecycle (Fig 4), an observation consistent with previous studies [66,76]. Not all *N. pachyderma* shells develop a secondary calcite crust with depth, and these thin non-encrusted shells can be found throughout the water column [66,77]. In the North Pacific, shell parameters of *G. bulloides* such as the area density and outermost chamber wall thickness increase 20% from the 0–50 m to the 100–150 m water depth interval [36]. We find similar results in the northern Barents Sea; the area density of *T. quinqueloba* increases by 50.1% from the 0–50 m to the 100–150 m water depth interval, while the mean area density of *N. pachyderma* increases by 29.5%. Furthermore, the CT numbers of *N. pachyderma* and *T. quinqueloba* increase by 10.2% and 20.3%, respectively, from the 0–50 m to the 150–200 m water depth interval. By the deepest sampling interval, 200–300 m, the CT number of *N. pachyderma* has increased by a further 6.6% (n = 32), resulting in a total increase in CT number by 15.8%. Below the 150–200 m water depth interval (n = 38), *T. quinqueloba* decrease in density by 3.4%. The shallower and narrower depth habitat in the water column of *T. quinqueloba* compared to *N. pachyderma* is reflected in the faster rate of both increasing shell density and shell thickening per meter. However, we find thin low-density shells and thick high-density shells of both species in the entire water column (Fig 2A and 2B). If we use thick high-density shells as a proxy for reproduction, then reproduction occurs in the entire water column. Cytoplasm-bearing specimens are also present in the entire water column (S7 Table in S1 File), although in lower abundance in the deepest sampling intervals, especially *T. quinqueloba*. The increasing density curve with water depth may partly be the result of a higher presence of dead shells that have already released gametes.

The decrease in the CT number of *T. quinqueloba* from the 150–200 m depth interval to the 200–300 m depth interval likely reflects the dissolution of their internal shell walls (S3 Fig). This internal dissolution may be due to gamete formation and release (Fig 3E), which has been documented to occur in certain PF species [67]. Early culture studies on PF also showed that dissolution starts in the internal shell walls [78,79]. In preparation for the release of gametes, PF increase the $\Omega_{Ca}$ of the microenvironment adjacent to their shell [80]. Some foraminifera may do so by discharging alkaline seawater vacuoles, which would result in the internal environment of the foraminifera to become less basic [81]. Another explanation for the internal dissolution is the oxidation of internal organic matter, documented in the pteropod species *Limacina retroversa* and *L. helicina antarctica* [82]. However, this is less likely in PF shells, because they are made of calcite, which is more robust than aragonite and the proportion of soft tissue to shell size is significantly smaller than in pteropods [83]. The $\Omega_{Ca}$ is supersaturated throughout the water column ($\Omega_{Ca}$ = 2.4–4), yet there are no known $\Omega_{Ca}$ thresholds for PF. The presence of *T. quinqueloba* shells in the deepest sampling interval may also reflect a relic population. The internally dissolved shells may have a slower sinking rate than the specimens without dissolved internal walls, making them more likely to be sampled.

At our study site, PF shell density is strongly related to shell volume (S4, S5 Tables in S1 File). In general, the larger the shell volume, the more dense it is. However, the increase in CT number with depth after size-normalization is still significant (p < 0.01). This means that the increase in shell density with depth is not a function of shell volume.

Our results highlight the importance of comparing PF in the same life stage, because the shell thickness and density gradually increases as they mature. The same size is not enough to eliminate ontogenetic effects (Figs 3 and 4), therefore it is also advisable to compare shells from the same sampling depth. In a study showing shell thinning in PF due to OA by comparing pre-industrial and modern shells, sampling depth may not have been the same [37]. A discrepancy in sampling depth may mean that the results simply show natural variation in shell thickness with depth.

The PF sampled from the water column in our study area did not show any signs of dissolution, both in the outer and inner shell wall (Fig 5). The only exceptions are some specimens of *T. quinqueloba* found below 150 m water depth (S3 Fig). There is a clear depth zonation in individual abundance [30], and shell density in both species. The increase in shell density with depth is in agreement with observations in the North Pacific [36], and is believed to be driven by ontogeny.

**4.1.2 Comparison *N. pachyderma* and *T. quinqueloba* from the sediment and species-specific dissolution.** The sedimentation rate in the northern Barents Sea ranges from 0.5–1.3 mm/yr [84], meaning that it takes anywhere from 8 to 20 years to accumulate 1 cm of sediment. The top 1 cm of sediments will therefore host PF that have settled at different times and thus can show a variable degree of dissolution (Figs 6 and 7). When PF from sediment samples are used in geochemical studies, it is often stated that the samples do not show any evidence of dissolution. The surface texture of *T. quinqueloba*, and especially that of *N. pachyderma* undergo only slight changes in their external appearance as they dissolve. The subtle dissolution in the surface texture may go undetected under a light microscope if all chambers are intact, which was the case for the samples used in this study. The post-depositional dissolution found in some of the specimens (Figs 6D, 6E and 7D–7F) is likely to alter the original chemical composition of their tests, mainly the Mg/Ca ratio, and the oxygen and carbon isotopic composition [85,86]. The higher percentage of low-density *N. pachyderma* shells (75%) compared to *T. quinqeloba* (39%) suggests that fewer low-density *T. quinqeuloba* shells remain intact in the surface sediments, which may lead to an underrepresentation of *T. quinqueloba* in the sediment records. Selective dissolution of *T. quinqueloba* is also likely because of the extensive internal dissolution in the low-density shells (Fig 6D and 6E), which could lead to a collapse of the entire shell resulting in fragmentation.

The inter-species differences in the manifestation of post-depositional dissolution is thought to be primarily due to the magnesium content in the calcite structure [87], thus also suggesting that the calcification process is species-specific. *Neogloboquadrina pachyderma* consistently rank as one of the planktonic foraminiferal species most resistant to dissolution, regardless of the region they are found, while *T. quinqueloba* has a low resistance to dissolution [87,88]. The exponential curve for *N. pachyderma* shell thickness versus CT number (Fig 5A) is steeper than that of *T. quinqueloba* (Fig 5B). The steeper *N. pachyderma* curve suggests that they calcify more than *T. quinqueloba*, leading to a thicker secondary crust. The ability to build a thicker and denser crust may have a number of different explanations. Firstly, there could be a difference in lifecycle length between *N. pachyderma* and *T. quinqueloba*. *Neogloboquadrina pachyderma* may have a longer lifecycle than *T. quinqueloba* meaning that they could calcify over a longer period of time and build thicker and denser shells. Individuals of *N. pachyderma* have been kept alive in culture for up to 200 days [89,90]. The tendency of *N. pachyderma* to build thicker and denser shells may be due to a naturally higher calcification rate, rather than a

longer lifecycle compared to *T. quinqueloba*. The two species may also have very different calcification strategies because, unlike *N. pachyderma*, *T. quinqueloba* builds numerous spines on most of its chambers at the expense of chamber walls resulting in thinner shells.

The two-layered dissolution pattern seen in *N. pachyderma* highlights their higher degree of resistance to dissolution (Fig 7F). A similar pattern was also found in *G. bulloides* [91]. The denser outer calcite of *G. bulloides* was resistant to dissolution and remained well preserved in water undersaturated with respect to calcite, while the Mg-rich inner calcite dissolved [91]. This mechanism of selective dissolution likely skews the sediment record to favor species with a dense outer calcite layer. Following dissolution in the surface sediments, the thickness of the shell walls remains intact while the whole shell gets a more porous crystalline structure, resulting in a lower mean CT number (Figs 6 and 7). In our study, the dissolved shells from the surface sediments plotted to the left of the trend line showcase this phenomenon (Fig 5A and 5B), suggesting that the comparison between CT number and shell thickness can be used as a tool to identify shells which have undergone either post-depositional dissolution or calcified in low $\Omega_{Ca}$ waters [92]. However, outliers may occur if specimens have an unusual morphology. A *T. quinqueloba* specimen with an abnormally large and low-density final chamber plotted significantly to the left of the other shells from the water column (Fig 5B). Large, yet low-density shells may be found when PF calcify in low $\Omega_{Ca}$ waters, and shift their ecological strategy to favor shell size over shell density [36], although, *T. quinqueloba* has been shown to present a large phenotypic variation related to changes in sea surface temperature [93].

### 4.2 *Limacina helicina*

**4.2.1 Distribution in the water column and shell density.**   In contrast to PF, *L. helicina* perform diel vertical migrations. Mature individuals diurnally migrate in the upper 200 m of the water column, while veligers and juveniles migrate in the top 50 m [94]. Like PF, it is also not known how the shell density of *L. helicina* changes with depth and increasing number of whorls. There is a skewness towards numerous small individuals at the surface, which is in agreement with previous findings in the polar region [95]. Because they migrate vertically, *L. helicina* showed less of a vertical zonation in shell density through the water column (S8-S10 Tables in S1 File). The statistical significance in the increase in shell density with depth is driven by the low-density, smaller specimens in the 0–50 m depth interval (S10 Table in S1 File). This is an observation consistent with their distribution in the water column [94]. The dominance of small individuals at the surface is likely because they have not developed their swimming wings and must therefore stay in the food-rich layer for growth. Once they have developed their wings they are able to migrate deeper in order to avoid predators, and this predation risk is likely what controls the vertical distribution of *Limacina helicina* [96].

**4.2.2 Dissolution of *L. helicina*, ontogeny, and future outlook.**   The connection between low $\Omega_{Ar}$ and shell damage in *L. helicina* has been confirmed by observations from marine environments with large natural gradients in the carbonate chemistry [97,98]. However, recent studies on the periostracum of *L. helicina* suggests that they may not be as sensitive to OA as previously claimed [99,100]. Further, an increased food supply may reduce or even negate the effects of living in low-$\Omega$ waters [101,102]. In the Arctic, *L. helicina* juveniles may experience waters with lowest $[CO_3^{2-}]$ and $\Omega_{Ar}$ during fall and winter, and it is unclear whether they calcify during this time or await elevated saturation states at the onset of $CO_2$ uptake by phytoplankton production in spring [18]. Seasonal decline in carbonate parameters was found to coincide with a higher proportion of pteropod shell dissolution in the North Sea [101]. *Limacina helicina* shell dissolution has been recorded at a $\Omega_{Ar}$ of 1.4 [97], and greatly reduced calcification at $\Omega_{Ar} < 1.2$ [102]. An $\Omega_{Ar}$ of 1.4 is close to the values we observe at the bottom waters

in our study area. Moreover, our saturation states are based on a summer situation when the surface water has higher saturation states than what we would expect in fall and winter.

The increase in the thickness of their shell apex with growth could mean that they are more resistant to dissolution if the $\Omega_{Ar}$ at out study site decreases in fall and winter (Fig 8D and 8E), and their depth habitat deepens with growth. In the surface water (0–50 m) during the summer, the $\Omega_{Ar}$ conditions are favourable (Fig 8B), allowing the small, low-density individuals to prioritize the growth of their muscles. Their thin and delicate shells during this stage of their life cycle will be less compromised with the higher $\Omega_{Ar}$. It is possible that the thickening of the shell apex with increasing whorl number could be linked to re-directing the energy to calcification after finalizing the development of their soft body. It has been demonstrated that *L. helicina* can add new shell material after damage [99], and as long as the $\Omega_{Ar}$ is $\geq 1.2$ ongoing thickening can occur over the entire shell, including the protoconch [102]. The repair mechanism of *L. helicina* and ongoing thickening means that they can choose specific areas of their shell to thicken after the initial calcification as part of a resilience strategy to environmental stress. Instances of over-calcification as a reaction to low $\Omega$ values have been found in barnacles [103] and coccolithophores [104,105], further suggesting that some calcifiers can re-direct energy for calcification when their shells are vulnerable. However, a study from an upwelling area in the northern California Current Ecosystem suggests that *L. helicina* produce thinner shells as an adaptation mechanism to lower $\Omega_{Ar}$ water [98].

Longer term studies using the techniques described here could shed light on the natural variability in the shell properties of *L. helicina* throughout their life cycle. Topics which could be addressed are to what extent calcification intensity varies with $\Omega_{Ar}$ and nutrients, and if specimens living in low $\Omega_{Ar}$ environments have adapted by building of thicker and denser shells. One could also investigate if there are geographical variations in whorl thickness depending on seasonality and chemical environment. Furthermore, with the ongoing climate change, water temperatures in the Barents Sea have increased [4] and are projected to continue to increase globally [106]. Synergistic effects of OA and warming have been demonstrated to be especially lethal for juvenile *L. helicina* [107,108], highlighting the need for a better understanding of the *L. helicina* calcification strategy.

## 5. Conclusions

The application of the XMCT scanning technique on the extant planktonic calcifying foraminiferal (PF) species *Neogloboquadrina pachyderma* and *Turborotalita quinqueloba* and the pteropod species *Limacina helicina* retrieved from stratified plankton net samples from the northern Barents Sea have provided us with a unique dataset to better understand the shell density distribution with depth and ontogeny of these species at high Arctic latitudes. We found that both PF and *L. helicina* increase in shell density with depth, however there were inter-species differences in the PF due to depth habitat and reproduction. *Neogloboquadrina pachyderma* tends to be both thicker and denser than *T. quinqueloba*, and continues to increase in density until the deepest sampling interval 200–300 m. *Turborotalita quinqueloba* decrease in shell density below the depth interval 150–200 m, this loss may be due to internal dissolution associated with gamete release or bacterial degradation of the cytoplasm. Our results highlight the importance of sampling at the same water depth interval when comparing PF calcification intensity. In the surface sediments (0–1 cm), the shell preservation state was highly variable in both planktonic foraminiferal species with little alteration of the surface shell texture or shell thickness. Only the average CT number that reflects the average shell density revealed that dissolution had occurred. In the surface sediments, *N. pachyderma* appeared more resilient towards post-depositional dissolution. In this area from the Barents Sea, the

living PF did not suffer from dissolution effects. Dissolution occurred after death and after settling on the sea floor. We observed that *L. helicina* thickens their shell apex as the number of whorls increase. There was a weaker zonation in shell density through the water column compared to PF, which is probably due to vertical migration. We recommend longer-term studies on planktonic calcifiers using the XMCT scanning technique. Longer studies in different carbonate chemistry environments would provide even greater insight on the natural variability in shell density. This knowledge is important in order to use PF and *L. helicina* as biological indicators for ocean acidification and to predict future developments in food webs. It is also important in the use of PF as paleo-proxies.

## Supporting information

**S1 Fig. Annotated *Limacina helicina* to demonstrate measurement of physical parameters.** Wall thickness measurements were done along a cross-section (blue), and diameter measured along white stippled line. Black circles show location of shell thickness measurements. The shell in the figure has 3.5 whorls. More details on whorl counting method can be found in Janssen [53].
(TIF)

**S2 Fig. Temperature and salinity profile at study area.**
(TIF)

**S3 Fig. Cross-sections of *Turborotalita quinqueloba* found in the 200–300 m water depth interval.** Scale bars measure 100 μm.
(TIF)

**S1 File.**
(XLSX)

## Acknowledgments

We thank the captain and crew of the R/V *Helmer Hanssen*, without whom this work would not have been possible. We thank Dr. Julie Meilland for giving early input which helped shape the manuscript, and Dr. Arunima Sen for assistance with statistical analysis. We would also like to thank Dr. Naomi Harada for her continued support of the collaboration between UiT and JAMSTEC. We are very grateful to Dr. Brett Metcalfe and two Anonymous reviewers and editor Dr. Lukas Jonkers for their comments that greatly helped us improve the manuscript.

## Author Contributions

**Conceptualization:** Siri Ofstad.

**Data curation:** Siri Ofstad.

**Formal analysis:** Siri Ofstad, Katsunori Kimoto.

**Funding acquisition:** Katsunori Kimoto, Melissa Chierici, Agneta Fransson, Tine Lander Rasmussen.

**Investigation:** Siri Ofstad, Katarzyna Zamelczyk, Katsunori Kimoto.

**Resources:** Katsunori Kimoto, Melissa Chierici, Agneta Fransson, Tine Lander Rasmussen.

**Supervision:** Katarzyna Zamelczyk, Tine Lander Rasmussen.

**Validation:** Katarzyna Zamelczyk, Katsunori Kimoto, Melissa Chierici, Agneta Fransson, Tine Lander Rasmussen.

**Visualization:** Siri Ofstad.

**Writing – original draft:** Siri Ofstad.

**Writing – review & editing:** Siri Ofstad, Katarzyna Zamelczyk, Katsunori Kimoto, Melissa Chierici, Agneta Fransson, Tine Lander Rasmussen.

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
