## [Decision Letter · Decision Letter 0]

22 Jul 2020

PONE-D-20-18293

Relationship between shell density and vertical distribution of living planktonic foraminifera and pteropod species *Limacina helicina* in the Barents Sea

PLOS ONE

Dear Dr.Ofstad,

Thank you for submitting your manuscript to PLOS ONE. After careful consideration, we feel that it has merit but does not fully meet PLOS ONE’s publication criteria as it currently stands. Therefore, we invite you to submit a revised version of the manuscript that addresses the points raised during the review process.

Please submit your revised manuscript as soon as possible. If you will need a long time  to complete your revisions, please reply to this message or contact the journal office at plosone@plos.org. Please include the following items when submitting your revised manuscript:

We look forward to receiving your revised manuscript.

Kind regards,

Gerald Ganssen

Academic Editor

PLOS ONE

Additional Editor Comments:

Dear authors, please find two reviews of your manuscript. Both reviewers have substantial and constructive criticism to you work. Please address all the comments carefully and explain in a separate file the reason if you do not agree.

We look forward to receive your revised manuscript in the near future.

With kind regards

Gerald

Journal Requirements:

3. We note that Figure 1 in your submission contain map images which may be copyrighted. All PLOS content is published under the Creative Commons Attribution License (CC BY 4.0), which means that the manuscript, images, and Supporting Information files will be freely available online, and any third party is permitted to access, download, copy, distribute, and use these materials in any way, even commercially, with proper attribution. For these reasons, we cannot publish previously copyrighted maps or satellite images created using proprietary data, such as Google software (Google Maps, Street View, and Earth). For more information, see our copyright guidelines: http://journals.plos.org/plosone/s/licenses-and-copyright.

3.1.    You may seek permission from the original copyright holder of Figure 1 to publish the content specifically under the CC BY 4.0 license.

3.2.    If you are unable to obtain permission from the original copyright holder to publish these figures under the CC BY 4.0 license or if the copyright holder’s requirements are incompatible with the CC BY 4.0 license, please either i) remove the figure or ii) supply a replacement figure that complies with the CC BY 4.0 license. Please check copyright information on all replacement figures and update the figure caption with source information. If applicable, please specify in the figure caption text when a figure is similar but not identical to the original image and is therefore for illustrative purposes only.

Reviewers' comments:

Reviewer's Responses to Questions

**Comments to the Author**

1. Is the manuscript technically sound, and do the data support the conclusions?

Reviewer #1: No

Reviewer #2: Yes

2. Has the statistical analysis been performed appropriately and rigorously? 

Reviewer #1: I Don't Know

Reviewer #2: Yes

3. Have the authors made all data underlying the findings in their manuscript fully available?

Reviewer #1: Yes

Reviewer #2: Yes

4. Is the manuscript presented in an intelligible fashion and written in standard English?

Reviewer #1: No

Reviewer #2: Yes

5. Review Comments to the Author

Reviewer #1: Review of Relationship between shell density and vertical distribution of living planktonic foraminifera and pteropod species Limacina helicina in the Barents Sea

This manuscript presents CT scans of planktonic foraminifera and pteropods from the Barents Sea. The title suggests that the study is focussed on morphological changes, namely shell density, through a vertical profile of the water column, however, the study site is only 340m water depth with strong lateral currents from the North Atlantic. It is not intuitive to me why the authors would have selected this location if the title were to reflect the original objective of this study.

There is no real story in this data. The data is poorly represented, in fact I’d go as far as to say misleadingly so, in order to derive any story at all. The story that is drawn, that pteropods adapt to ocean acidification by thickening their shell, is entirely unsubstantiated by the data presented here.

I summarise some of my issues below.

Introduction. I found the introduction to be poorly written and often ambiguous, leaving the reader guessing at what is meant… e.g. what is meant by “positive relationship” between ocean acidification and food? Moreover, the authors display no critical assessment of previous research. I would have liked the authors to show that they had considered the limitations or caveats of past studies, especially an appreciation for the limitations of experimental/incubation studies verses field observations, rather than implying that all studies are equally robust. Ocean acidification studies are full of inconsistencies and opposed methodologies. We can all learn from what has and has not worked in the past, it is a real shame that the authors did not take this opportunity to say why their study would make an advancement.

Method. For the foraminifera it is very clear that the orientation of the cross section, and how much of the inner shell walls are sampled, has an influence on the mean CT number. I suggest that this analysis could be more robust had the CT number only taken into consideration the shell wall of the penultimate chamber. i.e. inner shell walls may be reabsorbed during the forams life cycle and/or disproportionally sampled by the cross section. The final shell wall may still be growing. The penultimate shell wall, and only that part that has an outer surface, could arguably be a more consistent/robust measure than the mean of the entire cross section.

Results. I took the time to look at the excel file of data supplied as suppplement. I was unable to reconcile the values in the “estimate” columns in tables S4 and S5 with the raw data. What is “estimate”? Furthermore, the units in these tables do not make any sense… I am pretty sure that the shell thickness is not >40000 mm! I am not an expert in stats, but I know enough to understand that the stats do not really support the statement on line 280 or any of the following conclusions. The r2 values are nothing to write home about. I suspect that had a deeper water site been chosen, there may be more of a story in the vertical profile variability.

Line 285. It is revealed that up to 85 % of specimens recovered from >150m water depth were “not living”. There is no mention of these specimens being dealt with differently/isolated from statistical analysis. Rather, in line 291 and again in the discussion, statements like “gradual shell thickening with depth” with no further recognition of the fact that many/most specimens are dead and therefore their position in the water column is more likely to reflect settling out/reworking by lateral/upwelling currents and/or “reworking” by predators. Once you’ve revealed that the specimens are not living it becomes a nonsense to draw conclusions are the life stages/depth migration of these specimens.

Line 292+ Foram shell thickness of 2-4 microns? I suspect that is an order of magnitude out. Had the authors looked at other seminal works on N. pachyerma sinistral they would have known that there measurement of shell thickness was way out. Kohfeld and Kozdon references are amongst critical references omitted from this manuscript.

Line 295. What is �m-m?

Line 300. This is the ONLY parameter compared with shell thickness!

Line 324. What is this 2-layer dissolution pattern? I don’t see it in the figure.

Line 355. Because ��varies with depth!

Line 357. The authors conclude that shell thickness of L. helicina increases in sequential whorls. This is absolutely not supported by the CT images. i.e. the thickness of the shell whirls shown in figure 8f do not support the plot shown in 8d. The only part of the shell that shows anything like the thickening reported in the plots is the structural part of the shell at the central spiral which must increase with each whorl for structural reasons. There is absolutely no evidence to support thickening of the shell as an adaptation to ocean acidification. This is a poor attempt to drag a story out of this data. The discussion from line 483 onwards, based on these ill found conclusions, is completely unsubstantiated.

Reviewer #2: 1. Is the manuscript technically sound, and do the data support the conclusions?

Yes

2. Has the statistical analysis been performed appropriately and rigorously?

Yes. However, please include the statistics that relate to the following statement (line 42-44:’ We show that the density of L. helicina shell has an inverse relationship with ΩAr and hypothesize that the gradual thickening of the shell wall could be an adaptation against low Ω’)

3. Have the authors made all data underlying the findings in their manuscript fully available?

Yes. However, the authors should include in the supplement the net min, max, mid depth intervals environmental parameters (e.g., temperture, salinity, omega, etc) alongside what they have already included.

4. Is the manuscript presented in an intelligible fashion and written in standard English?

Yes

6. PLOS authors have the option to publish the peer review history of their article (what does this mean?). If published, this will include your full peer review and any attached files.

Reviewer #1: No

Reviewer #2: **Yes: **Brett Metcalfe

---

## [Author Response · Author response to Decision Letter 0]

29 Oct 2020

We thank both reviewers for their constructive comments and feedback. Both B. Metcalfe and anonymous reviewer #2 suggest major revisions. We have corrected our manuscript following the reviewer’s suggestions and address each comment specifically which we document in the file “Response to Reviewers”. In the few cases, where no change in response to a particular comment is made, an explanation is given.

We thank the anonymous reviewer#1 for pointing out some weaknesses in our title and introduction, which have helped us greatly in sharpening the introduction and making it clearer. We also thank the reviewer for pointing out important and useful references and showing us other unclear parts of the manuscript – which we have now taken care of in this revised version of the manuscript.

We thank reviewer #2, B. Metcalfe for encouraging more statistical analysis, thorough examination of the supplementary materials and comments pertaining to the figures. We also thank the reviewer for pointing out important and useful references which have strengthened the manuscript. 

We hope our responses are satisfactory and that the manuscript is more technically sound, clear and better written than the original version.

Yours sincerely,

Siri Ofstad, Katarzyna Zamelczyk, Katsunori Kimoto, Melissa Chierici, Agneta Fransson and Tine L. Rasmussen

---

## [Decision Letter · Decision Letter 1]

26 Jan 2021

PONE-D-20-18293R1

Shell density of planktonic foraminifera and pteropod species *Limacina helicina* in the Barents Sea: assessment of relationship to environment conditions

PLOS ONE

Dear Dr. Ofstad,

Thank you for submitting your manuscript to PLOS ONE. After careful consideration, we feel that it has merit but does not fully meet PLOS ONE’s publication criteria as it currently stands. Therefore, we invite you to submit a revised version of the manuscript that addresses the points raised during the review process.

ACADEMIC EDITOR: Please insert comments here and delete this placeholder text when finished. Be sure to:

Indicate which changes you require for acceptance versus which changes you recommendAddress any conflicts between the reviews so that it's clear which advice the authors should followProvide specific feedback from your evaluation of the manuscript

We look forward to receiving your revised manuscript.

Kind regards,

Lukas Jonkers

Academic Editor

PLOS ONE

Journal Requirements:

Additional Editor Comments (if provided):

Dear authors, please accept my apologies in the delay with handling this manuscript and the confusion related to reviewing the outdated version of your manuscript. The reviewer has now provided their comments on the revised version and suggests minor changes to the wording in the introduction and asks for some clarification. Please address these points, as well as those raised by the other reviewer in your revised manuscript. I look forward to receiving an updated version.

Reviewers' comments:

Reviewer's Responses to Questions

**Comments to the Author**

1. If the authors have adequately addressed your comments raised in a previous round of review and you feel that this manuscript is now acceptable for publication, you may indicate that here to bypass the “Comments to the Author” section, enter your conflict of interest statement in the “Confidential to Editor” section, and submit your "Accept" recommendation.

Reviewer #1: (No Response)

Reviewer #3: (No Response)

2. Is the manuscript technically sound, and do the data support the conclusions?

Reviewer #1: Partly

Reviewer #3: Yes

3. Has the statistical analysis been performed appropriately and rigorously? 

Reviewer #1: I Don't Know

Reviewer #3: Yes

4. Have the authors made all data underlying the findings in their manuscript fully available?

Reviewer #1: Yes

Reviewer #3: Yes

5. Is the manuscript presented in an intelligible fashion and written in standard English?

Reviewer #1: Yes

Reviewer #3: Yes

6. Review Comments to the Author

Reviewer #1: I do not have time to scrutinise the manuscript in detail, but have read through you responses to reviewers’ comments.

While most issues seem to have been acted on I must raise issue with the response to may final point regarding the shell thickness of L. helicina/Figure 8.

My observation that the only part of the shell to exhibit any variability in shell thickens was the spine was not a recommendation to measure shell thickness here! This would be the LAST place you should measure shell thickness for any environmental interpretation. The spine is structural and MUST increase in thickness as the diameter of the whorl it needs to support increases (in the same way that a tree trunk thickens are the tree gets taller). The outer wall is the area you should target and I maintain that these is no increase in thickness of the shell wall with increasing whorl number. This is worthy of note, but the spine thickness metrics and associated discussion is misguided and should be removed before this manuscript could be considered suitable for publication.

Reviewer #3: Comments for “Shell density of planktonic foraminifera and pteropod species Limacina helicina in the Barents Sea: assessment of relationship to environmental conditions”

I'm glad to have an opportunity to read a new manuscript to assess the shell density calcareous organism by MXCT scanning. It has been considered that the calcification of calcareous organism will be suffered from ocean acidification in the future, and number of studies tried to figure out the effect of ambient seawater acidification to their calcification intensity in the nature and culture experiment. In this study, authors focused on the two planktic foraminiferal species (T. quinqueloba and N. pachyderma) and a species of pteropoda (L. helicina) obtained from arctic sea, where there is concern about the ocean acidification progress. Using by sample set of plankton tow, they investigated the vertical distribution of shell density and its variance with migration under the present nature condition. Such basic information of variation in shell density with normal ontogenetic process is valuable for understanding the alternation of shell calcification condition due to external factor as like as ocean acidification in the following step of study. The research methodology and quality of data are fine, and I basically agree with their discussion. I believe that this manuscript is definitely suit for publishing from PLOS ONE. However, in order to make this manuscript clearer and easier to understand for reader, I’d like to suggest some points as follow.

Major comments

1. First, authors discussed about the vertical distribution of shell physical condition of living calcareous organisms against the background of the issue of ocean acidification. However, what they have done is the observation under the single environmental condition. Therefore, this study mainly reveals the growth process or ordinary life cycle of calcareous organisms that is ontogenetic process in other word. On the other hand, they did not compare the shell physical feature under the different acidification condition at different place or time. This means that it cannot discuss the impact of ocean acidification to calcification intensity of calcareous organisms from the sample set in this study. I suppose that author should clearly explain the importance for investigating the ontogenetic and growth process of these organisms. In my understanding, it would be like fine to add few sentences in Introduction that explain “what kind of knowledge is lacking in Ocean Acidification research” and “Why understanding ontogenetic and growth process of calcareous organism is necessary for Ocean Acidification research in the future”.

2. Second, I have something to worry about the shell samples from the surface sediments. In this study, authors also discuss about the shell samples from the surface sediment samples. However, I feel that this discussion is different from main theme of this study. I could not understand well why authors had to measure the shells in sediment sample. If they are definitely necessary in this study, authors should explain why it is necessary and what is the purpose of it. Furthermore, because some basic information of sediment samples is lack from this manuscript and figures, it is difficult to understand. The detail of samples can be quoted from other papers, but at least sediment samples location, water depth and super/under saturated to carbonate should be represented in the manuscript and/or figure of Map.

Other comments

Material and Methods

Line 128- 2.2 Sampling of marine calcifiers, and/or Figure 1 (Map)

I suggest to add more information about surface sediment sample (e.g., location, depth, etc).

Results

Line 236: The water column can be divided into two……

What is reason for separating the water column at 75 m? Is this same with thermocline? It is unclear for me how does this work in the following Results and Discussion.

Line 251-252: The outer shell walls are thick and dense, while…..

This is just a comment. This is very interesting result. To me, it seems to be caused by inorganic dissolution of juvenile shell after shell calcification. But I have no idea why the shell can be dissolved in the water column such quickly. If this is due to the process of gamtogenesis, it must be very energy consuming ontogenetic process for foraminifera, because it is time consuming process to dissolve the calcified shell in the nature condition.

Line 314-339: 3.3.2 Planktonic Foraminifera from the surface sediments

As describe in major comments, it is better to explain why measurement of shell samples from the surface sediment is necessary.

Line 360-361: The way that L. helicina is distribution in the water relationship…..

Is it possible that thinner shell wall is scanned as lower CT number due to the resolution of CT scanning? I don’t mind this in the case of planktic foraminifera, because they are enough thick, but I’m wondering what about the case of pteropod with very thin shell wall.

Discussion

Line 411-: Cytoplasm-bearing specimens are also present in the entire water column (S7 Table),

…

Were the shell samples divided into with/without cytoplasm under the CT scanning? I’d like to compare the CT images of shell with/without cytoplasm individually if possible.

7. PLOS authors have the option to publish the peer review history of their article (what does this mean?). If published, this will include your full peer review and any attached files.

Reviewer #1: No

Reviewer #3: No

---

## [Author Response · Author response to Decision Letter 1]

4 Feb 2021

We thank both reviewers for their constructive comments and feedback. Both anonymous reviewer #1 and anonymous reviewer #3 suggest minor revisions. We have corrected our manuscript following the reviewer’s suggestions and address each comment below. 

Yours sincerely,

Siri Ofstad, Katarzyna Zamelczyk, Katsunori Kimoto, Melissa Chierici, Agneta Fransson and Tine L. Rasmussen

Reviewer #1

I do not have time to scrutinise the manuscript in detail, but have read through you responses to reviewers’ comments.

While most issues seem to have been acted on I must raise issue with the response to may final point regarding the shell thickness of L. helicina/Figure 8.

My observation that the only part of the shell to exhibit any variability in shell thickens was the spine was not a recommendation to measure shell thickness here! This would be the LAST place you should measure shell thickness for any environmental interpretation. The spine is structural and MUST increase in thickness as the diameter of the whorl it needs to support increases (in the same way that a tree trunk thickens are the tree gets taller). The outer wall is the area you should target and I maintain that these is no increase in thickness of the shell wall with increasing whorl number. This is worthy of note, but the spine thickness metrics and associated discussion is misguided and should be removed before this manuscript could be considered suitable for publication.

Reply: We are glad our responses and actions regarding your previous comments are satisfactory. We attempted to make the discussion points on the thickening of the spine speculative but understand that it is controversial. Therefore, Figure 8 has been edited, all whorl thickness measurements at the spine have been removed along with all mentions of it in the manuscript including the methods, supplementary tables and Figure S3. 

Reviewer #3

Major comments 

1. First, authors discussed about the vertical distribution of shell physical condition of living calcareous organisms against the background of the issue of ocean acidification. However, what they have done is the observation under the single environmental condition. Therefore, this study mainly reveals the growth process or ordinary life cycle of calcareous organisms that is ontogenetic process in other word. On the other hand, they did not compare the shell physical feature under the different acidification condition at different place or time. This means that it cannot discuss the impact of ocean acidification to calcification intensity of calcareous organisms from the sample set in this study. I suppose that author should clearly explain the importance for investigating the ontogenetic and growth process of these organisms. In my understanding, it would be like fine to add few sentences in Introduction that explain “what kind of knowledge is lacking in Ocean Acidification research” and “Why understanding ontogenetic and growth process of calcareous organism is necessary for Ocean Acidification research in the future”.

Reply: Thank you for your in-depth comment on the overall objectives of the paper. We have now edited the introduction to make the research purposes clearer for the reader. As you suggested, we added a few sentences on why this research is relevant for ocean acidification research.

2. Second, I have something to worry about the shell samples from the surface sediments. In this study, authors also discuss about the shell samples from the surface sediment samples. However, I feel that this discussion is different from main theme of this study. I could not understand well why authors had to measure the shells in sediment sample. If they are definitely necessary in this study, authors should explain why it is necessary and what is the purpose of it. Furthermore, because some basic information of sediment samples is lack from this manuscript and figures, it is difficult to understand. The detail of samples can be quoted from other papers, but at least sediment samples location, water depth and super/under saturated to carbonate should be represented in the manuscript and/or figure of Map. 

Reply: We see the surface sediment samples as a continuation of the water column story which focuses on the ontogenetic processes. The ontogenetic processes will affect how well the foraminifera will be preserved in the surface sediments. This further demonstrates the importance in studying ontogenetic and growth process of calcareous organisms. Planktonic foraminifera from surface sediments are primarily used in paleostudies, and the shell condition influences geochemical measurements. The surface sediments samples show that not all planktonic foraminifera develop a crust after gametogenesis, and experience differing degrees of dissolution of the internal chambers. The differences in these ontogenetic processes not only has the potential to effect water column ocean acidification studies, but also paleo-studies. The purpose of the surface sediment samples to the manuscript has been made clearer in the introduction, and more information on the samples has been added to the Material and Methods chapter (see reply below).

Other comments 

Material and Methods 

Line 128- 2.2 Sampling of marine calcifiers, and/or Figure 1 (Map)

I suggest to add more information about surface sediment sample (e.g., location, depth, etc).

Reply: The coordinates, water depth and calcite saturation at the two box-core stations has been added to section 2.2.. The location is in the same area as the plankton tows, therefore no additional information has been added to the map in Figure 1. 

Results 

Line 236: The water column can be divided into two……

What is reason for separating the water column at 75 m? Is this same with thermocline? It is unclear for me how does this work in the following Results and Discussion.

Reply: No actual separation of the water column has been done during for instance, statistical analysis. The word “divided” has been removed in order to avoid confusion. You are correct, the thermocline is indeed located at 75 m water depth. This is mentioned in the results section for the water chemistry because in addition to the warmer water temperature in the 0-75 m layer, the Ω saturation is elevated. This detail is relevant when we discuss the specimens sampled in the 0-50 m layer. 

Line 251-252: The outer shell walls are thick and dense, while…..

This is just a comment. This is very interesting result. To me, it seems to be caused by inorganic dissolution of juvenile shell after shell calcification. But I have no idea why the shell can be dissolved in the water column such quickly. If this is due to the process of gamtogenesis, it must be very energy consuming ontogenetic process for foraminifera, because it is time consuming process to dissolve the calcified shell in the nature condition.

Reply: Thank you for your comment. Yes, this is a very interesting observation which could suggest that the internal environment is extremely corrosive following gametogenesis.

Line 314-339: 3.3.2 Planktonic Foraminifera from the surface sediments

As describe in major comments, it is better to explain why measurement of shell samples from the surface sediment is necessary.

Reply: The recently settled PF were primarily included to demonstrate the large range in calcite preservation in the 0-1 cm layer. We believe this will be of interest to the paleo-community. The preservation of the internal chambers will affect geochemical measurements, and the internal conditions were not apparent from the outer shell wall. Furthermore, by including the surface sediment samples we show an additional and promising use of the XMCT technology. As mentioned in the reply to your previous comment the reasoning for including the surface sediment samples has been made clearer in the introduction.

Line 360-361: The way that L. helicina is distribution in the water relationship…..

Is it possible that thinner shell wall is scanned as lower CT number due to the resolution of CT scanning? I don’t mind this in the case of planktic foraminifera, because they are enough thick, but I’m wondering what about the case of pteropod with very thin shell wall.

Reply: Thank you for this comment, this is an issue which is important to address. This misrepresentation of shell density in thinner material is referred to as the partial volume effect and is related to the spatial resolution and voxel number which constitute the shell wall. The partial volume effect appears at the boundary between the air and the (shell) material. Both ends of the shell is in contact with the surrounding air, and if the air has sufficiently larger volume than the material within one voxel, this voxel is drawn as low density. 

We agree that this could be an issue if using a low scanning resolution. Our analysis was performed with a sub-micron resolution (0.833 μm/voxel, due to size there were 5 pteropods out of 25 that were scanned with a resolution just over 1 μm). This resolution is sufficient to accurately separate the shell of the pteropod from the air that surrounds it, both internally and externally. Furthermore, with our scanning resolution we would only see this effect in shell material thinner than 1.7 μm (uses < 2 voxels). Such a thin shell wall is very limited to the internal chamber of the pteropod shell. In our study, the average shell thickness of pteropods was around 3 μm and is sufficient to accurately draw the shell wall, therefore we can say that the misrepresentation of thin shell walls as lower density is very small and/or negligible in this study.

Discussion 

Line 411-: Cytoplasm-bearing specimens are also present in the entire water column (S7 Table),

…

Were the shell samples divided into with/without cytoplasm under the CT scanning? I’d like to compare the CT images of shell with/without cytoplasm individually if possible.

Reply: Unfortunately, no comparison of the CT images of shells with and without cytoplasm was done for this study, but that is a point which we will include for future investigations.

---

## [Editor Report · Decision Letter 2]

11 Feb 2021

PONE-D-20-18293R2

Shell density of planktonic foraminifera and pteropod species *Limacina helicina* in the Barents Sea: assessment of relationship to environment conditions

PLOS ONE

Dear Dr. Ofstad,

Thank you for submitting your manuscript to PLOS ONE. Apologies again for the long delay in getting the correct reviews to you and thank you for your careful revision of your manuscript. I read it carefully and am happy to accept it after you have addressed some minor comments. The only reason why I recommend minor revision at this stage is to give you the opportunity to make some changes that I hope you agree will improve the manuscript.

- please introduce the concepts of migration (diel or ontogenetic) much earlier in the text, preferably in the introduction. They are both important for the interpretation of the results. Please also be more careful with the wording since it remains difficult to actually prove ontogenetic migration in foraminifera, it could also be that shells simply grow bigger/denser at a preferred depth.

- please explain better what is meant with secondary calcification and how it differs (or not) from encrustation.

- please explain earlier and in greater depth how habitat depth and (the pattern in) shell density could be related.

- the colour plots in figures 3, 4 and 8 are confusing because they suggest three dimensions (variability in the x direction), whereas I think they show the data from a single CTD profile.

I have also annotated the pdf, please also address the (minor) comments and suggestions there.

We look forward to receiving your revised manuscript.

Kind regards,

Lukas Jonkers

Academic Editor

PLOS ONE

---

## [Author Response · Author response to Decision Letter 2]

8 Mar 2021

Thank you for your comments and suggestions for further improvement of the manuscript. We have corrected our manuscript following your annotations in the PDF-file and address each of your four separate comments below. 

Yours sincerely,

Siri Ofstad, Katarzyna Zamelczyk, Katsunori Kimoto, Melissa Chierici, Agneta Fransson and Tine L. Rasmussen

------

- please introduce the concepts of migration (diel or ontogenetic) much earlier in the text, preferably in the introduction. They are both important for the interpretation of the results. Please also be more careful with the wording since it remains difficult to actually prove ontogenetic migration in foraminifera, it could also be that shells simply grow bigger/denser at a preferred depth.

Reply: Thank you for your comment, these concepts are central to the manuscript and therefore warrants a more in-depth explanation. We have now included the concepts of diel and ontogenetic migration in the introduction and have attempted to make the reader aware that ontogenetic migration is a disputed concept, but we are assuming that it occurs. 

- please explain better what is meant with secondary calcification and how it differs (or not) from encrustation.

Reply: Thank you for bringing this to our attention. Encrustation and secondary calcification are often used interchangeably in the literature, leading to some uncertainty (e.g. Kohfeld et al., 1996; Hillaire-Marcel et al., 2004). We have clarified what we mean by these terms (secondary calcification, gametogenic calcite addition and diagenetic encrustation) in the introduction, so it is clear to the reader what they refer to in this manuscript. We have also corrected instances where we have mixed the terms. 

- please explain earlier and in greater depth how habitat depth and (the pattern in) shell density could be related.

Reply: We have added this to the introduction following the introduction to diel and ontogenetic migration, and clearly state that we hypothesize that depth habitat and shell density are related, if we assume that calcification is continuous and ontogenetic vertical migration occurs. We have also added that we assume that denser shells found below lighter shells are older, as you pointed out in your annotation, we make this assumption later in the results and discussion.

- the colour plots in figures 3, 4 and 8 are confusing because they suggest three dimensions (variability in the x direction), whereas I think they show the data from a single CTD profile.

Reply: We wholeheartedly agree with this comment and have removed the aragonite and calcite background fill in Figures 3, 4 and 8. You are correct, the aragonite and calcite data are from a single CTD profile, therefore the profiles are now shown as an x and y plot.

---

## [Editor Report · Decision Letter 3]

15 Mar 2021

Shell density of planktonic foraminifera and pteropod species *Limacina helicina*  in the Barents Sea: relation to ontogeny and water chemistry

PONE-D-20-18293R3

Dear Dr. Ofstad,

We’re pleased to inform you that your manuscript has been judged scientifically suitable for publication and will be formally accepted for publication once it meets all outstanding technical requirements.

Kind regards,

Lukas Jonkers

Academic Editor

PLOS ONE

---

## [Editor Report · Acceptance letter]

16 Apr 2021

PONE-D-20-18293R3 

Shell density of planktonic foraminifera and pteropod species *Limacina helicina* in the Barents Sea:  relation to ontogeny and water chemistry 

Dear Dr. Ofstad:

I'm pleased to inform you that your manuscript has been deemed suitable for publication in PLOS ONE. Congratulations! Your manuscript is now with our production department. 

Kind regards, 

on behalf of

Dr. Lukas Jonkers 

Academic Editor

PLOS ONE